# The Genetic Diagnosis of Ultrarare DEEs: An Ongoing Challenge

**DOI:** 10.3390/genes13030500

**Published:** 2022-03-12

**Authors:** Luciana Musante, Paola Costa, Caterina Zanus, Flavio Faletra, Flora M. Murru, Anna M. Bianco, Martina La Bianca, Giulia Ragusa, Emmanouil Athanasakis, Adamo P. d’Adamo, Marco Carrozzi, Paolo Gasparini

**Affiliations:** 1Genetics Unit, Institute for Maternal and Child Health, Scientific Institute for Research, Hospitalization and Healthcare (IRCCS) Burlo Garofolo, 34137 Trieste, Italy; flavio.faletra@burlo.trieste.it (F.F.); annamonicarosaria.bianco@burlo.trieste.it (A.M.B.); martina.labianca@burlo.trieste.it (M.L.B.); giulia.ragusa@asufc.sanita.fvg.it (G.R.); emmanouil.a@gmail.com (E.A.); adamopio.dadamo@burlo.trieste.it (A.P.d.); paolo.gasparini@burlo.trieste.it (P.G.); 2Child Neuropsychiatry Unit, Institute for Maternal and Child Health, Scientific Institute for Research, Hospitalization and Healthcare (IRCCS) Burlo Garofolo, 34137 Trieste, Italy; paola.costa@burlo.trieste.it (P.C.); caterina.zanus@burlo.trieste.it (C.Z.); marco.carrozzi@burlo.trieste.it (M.C.); 3Pediatric Radiology Unit, Institute for Maternal and Child Health, Scientific Institute for Research, Hospitalization and Healthcare (IRCCS) Burlo Garofolo, 34137 Trieste, Italy; floramaria.murru@burlo.trieste.it

**Keywords:** neurodevelopmental disorders (NDDs), epileptic encephalopathies (EEs), developmental and epileptic encephalopathies (DEEs), whole-exome sequencing (WES), reverse phenotyping

## Abstract

Epileptic encephalopathies (EEs) and developmental and epileptic encephalopathies (DEEs) are a group of severe early-onset neurodevelopmental disorders (NDDs). In recent years, next-generation equencing (NGS) technologies enabled the discovery of numerous genes involved in these conditions. However, more than 50% of patients remained undiagnosed. A major obstacle lies in the high degree of genetic heterogeneity and the wide phenotypic variability that has characterized these disorders. Interpreting a large amount of NGS data is also a crucial challenge. This study describes a dynamic diagnostic procedure used to investigate 17 patients with DEE or EE with previous negative or inconclusive genetic testing by whole-exome sequencing (WES), leading to a definite diagnosis in about 59% of participants. Biallelic mutations caused most of the diagnosed cases (50%), and a pathogenic somatic mutation resulted in 10% of the subjects. The high diagnostic yield reached highlights the relevance of the scientific approach, the importance of the reverse phenotyping strategy, and the involvement of a dedicated multidisciplinary team. The study emphasizes the role of recessive and somatic variants, new genetic mechanisms, and the complexity of genotype–phenotype associations. In older patients, WES results could end invasive diagnostic procedures and allow a more accurate transition. Finally, an early pursued diagnosis is essential for comprehensive care of patients, precision approach, knowledge of prognosis, patient and family planning, and quality of life.

## 1. Introduction

The enormous flow of information resulting from the application of next-generation sequencing (NGS) techniques and the continued identification of new genes are changing the approach to early-onset epilepsies; the greatest success in discovering the epilepsy genes came from the study of epileptic encephalopathies (EEs) [1]. Genetic-driven data have revealed that genotype–phenotype correlations are complex and poorly defined [2], that the electroclinical and cognitive characteristics of specific genetic syndromes change during development and that the salient features may be less evident or even different in adulthood [3]. The increased knowledge of the etiology and underlying pathophysiological processes provided by genetic research have prompted clinicians to adopt different perspectives and nosological frameworks for accurate phenotypic characterizations and phenotype–genotype correlations [4,5]. In this regard, the term developmental and epileptic encephalopathies (DEEs) has been coined to define genetic wide electroclinical syndromes characterized by epilepsy, developmental delay or regression or intellectual disability, an abnormal EEG, and other possible neurological or systemic manifestations. An epileptic encephalopathy (EE), in which developmental delay or regression can be specifically attributed to the effect of an ongoing epileptic activity, can coexist and make the clinical picture and the therapeutical approach more complex [1]. There is a noteworthy clinical and genetic heterogeneity and sometimes overlapping features in these conditions, in particular in patients with early-onset epilepsy and intellectual disability (ID) [6,7] and in the wide field of neurodevelopmental disorders (NDDs). Heterogeneity, with multiple nonspecific features, suggests a genetic-first approach; therefore, whole-exome sequencing (WES), which covers the entire coding sequence, is becoming the current standard of care [8,9,10,11]. The literature emphasizes the “exome first” approach as the first-line test in the diagnostic pathway of individuals with NDDs and/or epilepsy and highlights the accelerating impact on diagnostic time intervals and reduced costs [12,13]. The currently achievable diagnostic yield of WES in DEE_S_ is 31–53% [14,15,16,17]. The difficulty deriving from the management of large amounts of WES data and from the interpretation of the pathogenic role of variants is one of the causes of undiagnosed patients [18,19,20,21]. The iterative and collaborative process between research and diagnostic laboratories, combining ongoing analysis with clinically oriented interpretation, can lead to a genetic diagnosis for a part of these subjects, including patients for whom clinical genetic tests had failed to produce definitive results [22,23]. Furthermore, increasing evidence suggests that a large percentage of NDD cases may be genetically complex and may occur through models involving mosaicisms, epigenetic mechanisms, and digenic/polygenic inheritance [24]. This complexity can confuse clinicians who rely on their diagnostic suspicion of the patient’s phenotypic characteristics and the available literature. Therefore, clinicians can be reluctant to move closer to an area that they still consider the prerogative of research primarily and postpone or abandon the goal of further investigation, thereby forgoing its potential benefits. Identifying the underlying genetic abnormality of EE/DEEs is critical as some are potentially treatable, and pharmacotherapy can be rationalized in a number of conditions [25]. In general, with the growing potential of precision medicine, it is essential to define the full range of the phenotypes that characterizes the natural history and evolution of DEEs and to identify appropriate clinical assessment tools [26]. There is also a rush to pursue the benefits of molecular diagnosis in older patients and in adults [27,28,29].

The study aims to (1) extensively illustrate the working method used to investigate patients with EEs, DEEs, and epilepsy with unspecific NDDs, including those whose previous clinical genetic tests were not conclusive, to encourage collaboration among health care providers, researchers, and clinical laboratories to reduce the gap between clinics and research; (2) highlight the dynamic and multidisciplinary nature of the diagnostic process, the importance of analysis and in some cases reanalysis of exome data, and the key role of reverse phenotyping in increasing the diagnostic yield; (3) discuss some topics, including alternative genetic inheritance patterns, in particular the contribution of autosomal recessive inheritance in outbred populations, the influence of biallelic pathogenic variants on clinical presentation, and the contribution of mosaicism; and (4) discuss the importance of the genetic diagnosis for overall patient care, family planning, and transition-specific issues.

In the following sections, we describe a dynamic exome analysis used to investigate 17 patients with EEs, DEEs, and epilepsy with unspecific NDDs and previous negative or inconclusive genetic testing. We report the molecular findings in the cohort and the detailed clinical description of the patients relevant to underline the relation between the working method and the diagnostic yield and introduce the discussion topics.

## 2. Materials and Methods

An overview of the diagnostic procedure adopted in this study is provided in Figure 1. A multidisciplinary team assessed the clinical presentation at disease onset and follow-up. The evaluation included electroclinical data, neuroimaging, cognitive and behavioral tests, dysmorphological features and anthropometric data acquisition, and metabolic and other diagnostic tests. We collected the family history, evaluated the relatives’ phenotyping, and, after genetic counseling, recruited the patients for molecular diagnostic purposes. The individuals, negative for pathogenic micro-rearrangements (SNP array negative), were enrolled for WES analysis. Subsequently, WES data and the phenotypic information were discussed in interdisciplinary meetings; the interpretation of WES data was supported by systematic bibliographic review and public database consultation. All selected variants were confirmed, and segregation analysis was performed with Sanger sequencing. A reverse phenotyping strategy was used to refine the phenotype based on the interpretation of genetic data [18].

### 2.1. Study Design and Participants

The study involved 17 individuals with EEs, DEEs, and epilepsy with unspecific NDDs, referred by the Neuropediatrics Department at IRCCS Burlo Garofolo (Trieste, Italy), without a molecular diagnosis detected on previous genetic tests and therefore recruited to perform diagnostic WES analysis. The detailed clinical information was systematically gathered at the initial presentation and during a multidisciplinary team’s follow-up. The clinical information was standardized using the Human Phenotype Ontology (HPO) [30].

The inclusion criteria were (1) the developmental slowing, regression, or plateauing occurring on a background of normal development and emerging with seizure onset and/or frequent epileptic activity on the EEG; (2) the “pre-existing developmental delay”, complicated by plateauing or regression concurrently with seizure onset and/or frequent epileptic activity on the EEG; (3) drug-resistant epilepsy for a prolonged period; and (4) the availability of DNA of both parents. The exclusion criteria were (1) metabolic abnormalities; (2) vascular stroke, head injury, ischemia, or infections; and (3) abnormalities detected on previous genetic tests, including the presence of micro-arrangements revealed with high-resolution molecular cytogenetics methods.

Genomic DNA was extracted from patients, parents, and siblings, when available, from venous peripheral blood lymphocytes according to standard procedures. Investigations were conducted according to the Declaration of Helsinki principles. Informed consent from the enrolled subjects was collected.

### 2.2. WES, Interpretation, and Validation

WES was performed on the trio (patient and both parents) to identify a molecular diagnosis. In brief, genomic DNA was processed for library enrichment according to the Twist Human Core Exome Kit v.1.3 (Twist Bioscience). Sequencing was performed on an Illumina Nextseq500 or Novaseq 6000 (Illumina) system with an average depth on target of 104X, with ≥99.4% of coding exons sequenced to a > 20X (Appendix A). The Burrows-Wheeler Aligner (BWA 0.7.17) was used to align 150 bp pair-end sequencing reads to the human reference genome (GRCh38/hg38). Variant calling was conducted employing Genome Analysis Toolkit (GATK 4.1). Annotation and prioritization of the variants were accomplished based on the American College of Medical Genetics and Genomics (ACMG) guideline [31]. SNVs and INDELs were filtered using ANNOVAR software referring to several public databases (dbSNP build150, NHLBI Exome Sequencing Project (ESP), Exome Variant Server, Genome Aggregation Database (gnomAD)) and led to ruling out those variants previously reported as polymorphism. In particular, a minor allele frequency (MAF) cut-off of ≤0.01% was used. All trios were examined for the following inheritance patterns: *de novo* dominant, homozygous recessive, compound heterozygous, and hemizygous.

The pathogenicity of known genetic variants was assessed with the ClinVar, The Human Gene Mutation Database (HGMD), Online Mendelian Inheritance in Man (OMIM), and DECIPHER. Several in silico tools, such as PolyPhen-2 [32], Sorting Intolerant from Tolerant (SIFT) [33], MutationTaster [34], and Combined Annotation Dependent Depletion (CADD) score [35], were applied to establish the pathogenicity of novel variants. Moreover, the evolutionary conservation of residues across species was evaluated by Genomic Evolutionary Rate Profiling (GERP) score [36]. regSNPs-splicing was adopted to predict the effect of the splice site mutations [37].

Finally, on a patient-by-patient basis, identified variants were discussed in the context of phenotypic data at interdisciplinary meetings. The most likely disease-causing SNVs/INDELs were analyzed by direct Sanger sequencing, which was also used to investigate the segregation analysis within the family. We described the molecular diagnosis as “definite” based on our overall clinical assessment of whether the variant(s) explained the clinical features, taking into account the ACMG classification, the inheritance pattern, clinical (full or partial) fit between the patient’s HPO terms and the reported clinical phenotypes for the genes or variants. We considered “possible” a case that presented already reported variants in a disease gene but presented a phenotype that did not entirely fit the disorder’s known clinical presentation.

## 3. Results

### 3.1. Cohort

The study involved 17 index cases (9 males and 8 females), mostly sporadic, with EEs, DEEs, and epilepsy with unspecific NDDs which met the inclusion criteria. One had an affected sibling. Among participants, parental history of consanguinity was not reported or known. The mean patient age at enrollment was 14.8 ± 5.5 years (Table 1) with seizure onset between birth and 6 years of age. Eleven patients were over 15 years old, five of whom were over 20 at their study enrollment. Family history was unremarkable in all but two; in particular, patient 2 has an older affected brother with similar clinical presentation, and the father of patient 3 reported having seizures in childhood.

In all the probands, diagnostic investigations had been initiated during childhood with regular follow-up. In most of them, epilepsy had arisen in the first three years of life (26 months on average). Patients took on average of 9.5 different antiseizure medications (ASMs) throughout their disease, and seizures were refractory in about 59% of patients at the last assessment (Table 2 and Appendix A). Metabolic abnormalities and multiorgan systemic anomalies were absent in the patients included in the study. All presented with epilepsy, and the majority had a severe phenotype with developmental delay (DD) or ID (88%), neurological speech impairment (82%), microcephaly (24%), and abnormal muscle tone (53%). Autistic spectrum disorder features were presented in 18% of the probands. Major clinical characteristics of the patients are summarized in Table 2.

Patients underwent biochemical testing and a combination of clinical genetics testing, including specific genes and/or comprehensive gene panels for most common EE genes, without receiving a diagnosis. All patients had molecular karyotype to exclude chromosomal deletions or duplications, which may account for 5–10% of cases [38].

### 3.2. Diagnostic Rate of WES

Overall, a definite molecular diagnosis with variants in well or newly established disease genes was provided for 10 out of 17 cases (58.8%). In addition, 1 out of 17 (5.9%) had a possible molecular diagnosis. About 12% of cases presented variants of uncertain significance (VUS) in candidate disease genes that need further confirmation, and 23.5% of the cases remained unsolved (Table 3).

Five out of ten individuals with a definite diagnosis (50%) had variants inherited in an autosomal recessive manner, including three compound heterozygous genotypes and two homozygous variants. Four cases (40%) had a *de novo* variant. A pathogenic somatic mutation was identified in one case, and 46% were novel variants. Variant types included 10 missense variants, 2 nonsense mutations, and a canonical splice site variant (Table 4).

### 3.3. Case Presentation and WES Results

Clinical details of all patients with the defined and possible diagnoses are presented in Appendix A, and WES results are presented in Table 4. The homozygous mutation in *ROGDI* carried by case 2 was also confirmed in the older affected brother, and the pathogenic variant in *GNB1* identified in patient 7 was previously reported; their phenotypes are consistent with those described in the literature [43,44]. During the finalization of the study and manuscript preparation, patients 1, 8, 9, and 10 were eventually presented in a frame of more extensive gene-specific cohort description. These results allowed us to confirm a specific electroclinical syndrome (*SLC13A5*), to add new elements to known DEEs or NDDs (*SPATA5*, *CSNK2B*), and to substantiate a candidate gene as a disease gene (*SPEN*) [39,40,41,42].

Cases relevant to underline the relation between the working method and the diagnostic yield and introduce the discussion topics are detailed below.

#### 3.3.1. Case 3

Case 3 was a 20-year-old boy at last examination. His father suffered from seizures during a short period of his infancy. The proband’s psychomotor and cognitive development was utterly normal until epilepsy started at 3 years with an electroclinical picture of typical absences of infancy. Brain MRI was normal. Pharmacoresistant myoclonic, atonic, generalized tonic–clonic seizures ensued. The boy presented recurrent respiratory infections that resulted in a worsening in seizure control with the occurrence of repeated myoclonic status epilepticus that, on one occasion, needed prolonged ICU hospitalization. A progressive cognitive decline became evident, with severe dysarthria, dysphagia, and an ataxic gait. At last examination, MRI showed a mild cortical and subcortical atrophy, and the EEG revealed a global slowing and diffuse, bifrontal epileptic activity.

WES analysis identified compound heterozygous variants affecting the *CACNA1H* gene, which encodes for the α1 subunit of a voltage-sensitive Ca (2+) channel. Susceptibility to childhood absence epilepsy-6 (ECA6) (MIM: #611942) and idiopathic generalized epilepsy-6 (EIG6) (MIM: #611942) are conferred by heterozygous variations in this gene. The missense mutation c.G3175T, p.Ala1059Ser (NM_021098) was inherited from the father, who reported having seizures in childhood that resolved spontaneously. The second change identified in our patient was a novel variant (c.C2329T, p.Arg777Cys) in exon 10, which was inherited from the healthy mother. The mutation was predicted to be damaging using several in silico prediction tools and a CADD score of 23.5. Based on the nature and location of the identified amino acid changes, a cumulative and detrimental effect was hypothesized.

#### 3.3.2. Case 4

Case 4 was a 12-year-old girl at last examination. The girl was born from a twin biplacental pregnancy complicated at 36 weeks of gestation by the onset of a maternal convulsive seizure, which resulted in fetal distress and emergency cesarean section. At birth, the girl presented congenital microcephaly and signs of not definitely confirmed prenatal CMV infection. A developmental delay emerged in the first year of life. Brain MRI showed patchy demyelination of subcortical white matter with ventriculomegaly without calcifications. The first seizure was reported at 23 months during fever, followed by a second episode in apyrexia a few months later; then, rapidly, the clinical picture evolved to a pharmacoresistant severe epileptic encephalopathy with daily seizures. The EEG recordings documented the recurrence of very high frequency critical events in wakefulness and during sleep and the presence of continuous paroxysmal activity in sleep.

At last examination, epilepsy, spastic/dystonic tetraparesis, absence of speech, and severe cognitive deficit constituted the clinical picture. WES analysis identified compound heterozygous mutations in the *SCN1A* gene. *SCN1A* encodes for the α subunit of a voltage-dependent sodium channel, a heteromeric complex (one glycosylated α subunit and two smaller β subunits), which regulates sodium exchange between intracellular and extracellular spaces and is essential for the generation and propagation of action potentials in neurons. *SCN1A* heterozygous mutations are associated with a broad phenotypic spectrum of epilepsies ranging from genetic epilepsy with febrile seizures plus (GEFS+) to DEEs [45].

The patient inherited a c.C3521G (p.Thr1174Ser) from the healthy mother and a c.C5782G (p.Arg1928Gly) from the healthy father. Both mutations have been reported previously (HGMD: CM076496 and CM081420, respectively) in the heterozygous state, the first enriched in patients with myoclonic epilepsy and GEF+ and the second in severe myoclonic epilepsy of infancy [46], and have allele frequencies in the general population of 0.001301 and 0.001706, respectively. The mutations were confirmed by Sanger sequencing and cosegregated with the disease in the family. The unaffected twin sister resulted wild type for both variants.

Based on the nature and location of the identified amino acid changes, a cumulative and detrimental effect was hypothesized.

#### 3.3.3. Case 5

Case 5 was a 21-year-old boy at the last examination. Focal onset to bilateral tonic–clonic seizures appeared at 8 months of life; the clinical presentation was complicated by atypical absences from the age of 5 and generalized tonic seizures from 7. Epilepsy did not respond to treatments. The interictal EEG evidenced generalized slow and paroxysmal activity (PO 2.5 Hz). Brain MRI, performed at 14 years of age, revealed an enlarged cisterna magna. The boy showed a progressive neurological decline. At last examination, epilepsy, moderate intellectual disability, hyperactivity, and self-directed aggressive behavior constituted the clinical picture. The patient harbored a *de novo* heterozygous mutation in *CDKL5*. Mutations in this gene are associated with the X-linked very severe developmental and epileptic encephalopathy-2 (MIM: #300672). The c.100-1G > A mutation, classified as pathogenetic (ACMG) [31] and never described before, affected a canonical splice site and was expected to induce a splicing change. Sanger sequencing confirmed the *de novo* origin of the mutation, and chromosome analysis revealed a 46, XY karyotype supporting somatic mosaicism for the c.100-1G > A.

#### 3.3.4. Case 6

Case 6 was a 4-year-old girl at last examination. The girl had a psychomotor delay, and at 11 months of age, epileptic spasms, focal seizures, and EEG disorganization with multifocal epileptic activity were diagnosed, configuring a DEE. The epileptic encephalopathy was not affected by ASM and repeated steroid cycles. The evolution of the neurological picture was characterized by severe psychomotor delay, aposturality, diffuse hypotonia, absence of language, neurosensory hearing loss, and microcephaly. She showed hyporeflexia with average nerve conduction velocity (VCN) and mild hyperlactatemia. Brain MRI showed a “Leigh-like” picture, with supra- and subtentorial cortical atrophy and hyperintensity of the basal ganglia.

WES analysis identified a *de novo* pathogenic missense mutation (NM_001303256: c.G79A, p.Glu27Lys) in exon 2 of the *MORC2* gene encoding a member of the ATPase family essential for epigenetic silencing through chromatin modification. It regulates the condensation of heterochromatin in response to DNA damage and plays a role in repressing transcription [47].

At the time of discovery, the variant identified in the patient was only described in a child with developmental delay from the DDD study, but no detailed clinical features were available [48]. Mutations in the *MORC2* gene were only associated with autosomal dominant sensory–motor neuropathy type 2z (CMT2Z, MIM: #616688), which clinically did not explain the phenotype in the patient. The review of genetic data over time has allowed finding a very recent study in which, using a hypothesis-free approach, *de novo*
*MORC2* mutations were identified in a cohort of patients with neurodevelopment disorder. Significantly, p.Glu27Lys has been described in five patients, including this case, affected by developmental delay, growth retardation, brain abnormalities, and inconsistent neuromuscular complaints [49].

#### 3.3.5. Case 11

Case 11 was a 14-year-old boy at last examination. Focal myoclonic seizures appeared at 27 months, initially clustering during somnolence and light sleep. The EEG evidenced an electrical status epilepticus in sleep that proved to be refractory to treatment. Seizure-free periods (from a few weeks to a few months) alternated with seizure clusters and tonic seizures during sleep or awakening until 9 years. At the last examination, clinical seizures were absent, but the EEG showed intense paroxysmal activity during sleep. The boy had a severe intellectual disability with autistic features and behavioral problems. The neurological examination was normal. Brain MRI revealed unspecific cerebellar abnormalities (mild vermis hypoplasia, retrocerebellar arachnoid cyst). WES unveiled a compound heterozygous genotype of the *SACS* gene, which comprised a missense mutation on the maternal allele (NM_014363: c.G2983T, p.Val995Phe) and a second one on the paternal allele (NM_014363: c.C3427A, p.Gln1143Lys). The mutations cosegregated with the disease in the family. Mutations in *SACS* cause autosomal recessive spastic ataxia of the Charlevoix–Saguenay type, the second most common cause of recessive ataxia, with >300 mutations described worldwide according to HGMD.

With an increasing number of reported patients, it became evident that atypical clinical cases existed, some of them without spasticity or peripheral neuropathy and the classical triad of symptoms [50,51,52].

The variants identified in our patient have been described previously; in particular, the p.Val995Phe was noted in a patient with spastic ataxia (cerebellar and pyramidal signs) together with a different mutation (p.Thr2529Ile) [53] and reported as a variant of uncertain significance in HGMD database. The p.Gln1143Lys in trans with p.Ala3661Val was detected in a patient with spastic paraplegia [54] as the disease-causing variant (HGMD).

## 4. Discussion

Our study identified molecular diagnosis in around 59% of participants, a yield higher than a range of 30% to 50% of DEEs attributed to a pathogenic variant [55,56]. As recently suggested, these results confirmed that the probability of reaching a correct diagnosis was enhanced by the methodological approach, particularly by in-depth phenotyping and the reanalysis of previous negative or uncertain sequencing data [18].

### 4.1. Alternative Genetic Inheritance Patterns. The Contribution of Autosomal Recessive Inheritance in an Outbred Population

In our series, we confirmed the contribution of alternative genetic inheritance patterns. In particular, an autosomal recessive pattern of inheritance was present in six cases (patients 1, 2, 3, 4, 8, and 11). Although most DEE genes harbored *de novo* pathogenic variants, the contribution of autosomal recessive inheritance in the outbred population is likely underestimated. Initially, it was considered rare because most early genetic discoveries were detected in consanguineous populations, but its relevance is likely to grow as it showed an average yield of 13% to 40% in previous studies [57,58] and reached a 54.5% yield in our cohort.

### 4.2. The Influence of Biallelic Pathogenic Variants on the Clinical Presentation

Increasing evidence indicates that a subset of genes that usually harbor *de novo* changes are associated with biallelic pathogenic variants. The clinical presentation of these individuals was often more severe than that of cases with *de novo* variants; examples were the *CACNA1A* and *SCN1B* mutations, in which the increase in severity was probably due to the complete absence of functional channels compared to haploinsufficiency in the case of *de novo* variants [59].

In this regard, we identified a patient with a very severe phenotype and compound heterozygous changes in *SCN1A* (Table 4).

Patient 4 is a girl with a severe early/onset epileptic encephalopathy and compound heterozygous *SCN1A* missense variants (p.Thr1174Ser and p.Arg1928Gly). Mutations in this gene, encoding the α1 subunit of the voltage-gated sodium channel, are associated with a wide range of disorders, including genetic epilepsy with GEFS+, familial hemiplegic migraine (FHM), Dravet syndrome (DS), and a form of very severe early infantile encephalopathy [45]. Hundreds of sequence variants of the *SCN1A* gene have been identified; most were *de novo*, and a few were inherited with carriers with or without mild clinical features. Recently, Brunklaus and colleagues described two families with a recessive pattern of inheritance and febrile seizures plus or Dravet syndrome, with a marked intrafamilial variability and heterozygous carriers with no symptoms [60]. Predicting disease outcomes based on variant type remains challenging [61]. The location of an *SCN1A* missense change in a less crucial functional area of the protein or less conserved physicochemical properties of the involved amino acid could allow heterozygous carriers not to be affected; changes on both alleles could have a cumulative and detrimental effect. The *SCN1A* Arg1648His mouse model was an impressive example of the worsening effect of homozygous mutations. Although heterozygous animals have an average lifespan and rare generalized seizures as adults, homozygous mice showed spontaneous generalized seizures and a shortened half-life [62]. Notably, more than 40% of the *SCN1A* variants reported in Clinvar have been classified as VUS, indicating that the functional characterization and, in this specific case, the cocharacterization of two variants is required.

### 4.3. The Role of Recessive Mutations in “Susceptibility” Genes

The *CACNA1H* gene encodes the pore-forming α1 subunit of the T-type calcium channel isoform Ca_v_3.2. Despite having been reported animal models carrying *Cacna1h* mutations that cosegregate with epilepsy phenotypes [63], in humans heterozygous mutations have been only associated with susceptibility to generalized epilepsy and focal or multifocal epilepsy and DEEs of varying severity [64,65]. In line with these findings, many *CACNA1H* variants are reported in healthy databases (GnomAD, Exac), suggesting that their contribution to human epilepsy may be low or might be dependent on additional genetic and/or environmental factors.

Patient 3, presenting with severe epileptic encephalopathy, carries two variants in *CACNA1H*: (1) c.G3175T (p.Ala1059Ser) located in a hotspot region for the action of the regulatory pathway [66] inherited from his father, reported to have seizures in childhood which resolved spontaneously, and (2) the rare c.C2329T, p.Arg777Cys variant inherited from the healthy mother and never reported in homozygous state.

The Ala1059Ser is already known as a susceptibility variant in individuals with different generalized epilepsies, as demonstrated by electrophysiological experiments [67]. Although none of the variants found in the human *CACNA1H* gene so far are sufficiently pathogenic to cause epilepsy on their own [68], a change may act in combination with other variants or environmental factors to force the level of neuronal excitability above the seizure threshold.

In this regard, a patient with absence epilepsy and body-wide chronic pain has been reported to have two inherited *CACNA1H* variants, p.Ala1059Ser and Pro769Leu [69]. The authors demonstrated that the coexpression of both variants caused an additive impairment on channel activity [69]. Interestingly, the girl showed symptoms after a viral infection, suggesting a trigger effect. Indeed it is noteworthy that the Cav3.2 channel is upregulated in response to inflammation [70].

In this respect, patient 3 showed a worsening seizure control after respiratory infections. These episodes resulted in the occurrence of repeated myoclonic status epilepticus that, on one occasion, needed prolonged ICU hospitalization.

Moreover, an inherited *CACNA1H* variant has already been reported in worsening a DEE phenotype in association with another gene [71].

In this view, although electrophysiological studies would be required to clarify the detailed pathogenetic mechanism of *CACNA1H* biallelic variants, these current findings might shed light on the impact of rare ion channel variants on the etiology of EEs.

### 4.4. The Contribution of Mosaicism

In case 5, we identified a novel heterozygous mutation in *CDKL5*. Mutations in this gene are associated with the X-linked very severe developmental and epileptic encephalopathy-2 (MIM: #300672). In large cohorts of boys with early-onset EE, *CDKL5* mutations accounted for about 3–5.4% [72]. In addition, increasing evidence indicates that somatic mosaicism for *CDKL5* mutations in boys with early onset is likely to be more frequent than previously ascertained, probably because their less deleterious effect enhances the viability of the male embryo [73,74,75]. Fifty males with *CDKL5* have been reported so far. The type and position of *CDKL5* variants with different impacts on the protein influence the clinical presentation. In males, postzygotic mosaicism, which accounts for 16% of the cases, contributes to this variability. Based on these issues, genotype–phenotype correlations are still challenging [76].

### 4.5. Latest Genetic Discoveries May Challenge Syndromic Classification Systems: The Example of KTZ Syndrome

We found two patients (cases 1 and 2), the first with a homozygous mutation in *ROGDI* (c.C286T; p.Gln96X), and the second with compound heterozygous mutations in *SLC13A5* (c.C1421T, p.Pro474Leu; c.G655A, p.Gly219Arg) (Table 4 and Appendix A). The combination of epileptic encephalopathy, developmental delay or regression, and yellowish discoloration of the teeth due to amelogenesis imperfecta (AI) was first described by Kohlschütter and colleagues in 1974 in a large Swiss family and is now recognized as Kohlschütter–Tonz syndrome (KTZS, MIM: #226750). In 2012, biallelic mutations in *ROGDI* (MIM: #614574) were identified as the cause of KTZS in most families, confirming autosomal recessive inheritance [77]. In 2016, the same authors reported that *ROGDI*-negative individuals with the clinical diagnosis of KTZS showed biallelic mutations in *SLC13A5*, previously described in individuals with the diagnosis of autosomal recessive early infantile epileptic encephalopathy (EIEE25, MIM: #615905) who also displayed variable teeth hypoplasia and/or hypodontia [78].

An in-depth analysis of our two patients’ extended clinical follow-up and EEG, associated with a comprehensive literature review, highlighted and confirmed the specific distinctive elements associated with the two genetic causes, including the age of seizure onset, the evolution of epilepsy and encephalopathy, and dental involvement with clinical and histological peculiarities, emphasizing that under the eponym KTZS coexist two syndromes (making the KTZS eponym obsolete).

### 4.6. When the “Reverse Phenotyping Approach” Is a Challenge. The Uncertainty of Interpreting the Clinical Contribution of Variants Associated with a Disease That Does Not Correlate with the Specific Phenotype. The Case of SACS

Over 170 *SACS* mutations with diverse phenotypes have been reported worldwide and thought to cause loss of function of sacsin. Reports described patients with atypical features, ataxia, and peripheral neuropathy, which included delayed-onset ataxia; nonataxic spastic paraplegia; mild pyramidal signs; cognitive decline; widespread supratentorial brain abnormalities; and episodic conditions such as epilepsy, including progressive myoclonus epilepsies and paroxysmal kinesigenic dyskinesia [79].

The exact frequency of these atypical clinical manifestations is complicated to determine at this point because most of the studies focused mainly on the predominant movement disorder, and other manifestations were poorly described. However, with the identification of patients with homozygous or compound heterozygous *SACS* mutations worldwide, it is now evident that the clinical features can vary widely, some of them lacking spasticity or peripheral neuropathy, and nonmotor symptoms such as cognitive decline, ID, and behavioral abnormalities can be prominent and even dominant clinical features [51].

Case 11 was a 14-year-old boy with two known *SACS* mutations and a clinical picture of ID, autism, epileptic encephalopathy. The classic triad of signs, including cerebellar ataxia, spasticity, and peripheral sensorimotor neuropathy, was not present at the last examination, but a wide range in age of onset of the symptomatology was described. The patient’s MRI revealed a volume loss within the superior components of the cerebellar vermis reported in 60.8% of the cases. The bilateral and symmetrical linear hypointensities previously identified in 33.3% of all affected subjects in the pons on T2 and T2-FLAIR sequences [80] were absent in our case.

### 4.7. The Importance of Revisiting a Patient’s Genetic Data over Time

Mutations in *MORC2* were first reported in 2016 in association with a progressive axonal and sensory neuropathy frequently presenting in the first decade of life [81]. Subsequently, few severely affected individuals were reported, although with inconsistent features. Using a genotype-first approach, in 2020, Guillen Sacoto and colleagues identified *de novo*
*MORC2* variants in patients with a severe neurodevelopmental disorder characterized by global developmental delay, short stature, microcephaly, and variable dysmorphic facies, with or without neuropathy and features of Leigh-like syndrome [49]. Reanalyzing after a given interval of time, in the light of knowledge that increases with incredible speed, has allowed us to interpret patients’ genetic data.

### 4.8. The Importance of a Genetic Diagnosis during the Transition

The age of transition is a vulnerable period for patients with complex comorbidities, who have to leave a coordinated and holistic pediatric approach for adult care. Identifying genetic etiology among those patients can be challenging for the adult neurologist, as the long history of seizures and complex antiseizure therapy can falsify their syndromic features, it is frequently difficult to reconstruct an accurate clinical history from the family, essential details may be missing, and medical records are often not available. Moreover, retrieving blood samples of older family members can be particularly difficult in adults [82].

Receiving a genetic diagnosis during transition can be a milestone providing reference points to the patient’s family for access to disease-dedicated associations and to the adult neurologist for organizing medical care and surveillance of disease-related conditions. In our cohort, the mean age of patients at which definite or possible genetic diagnosis was reached was 14.9 ± 6.2 years.

In these subjects, the previous clinical genetics evaluation and combination of genetic tests (specific genes and/or comprehensive gene panels for most common genes responsible for EE) varied as a part of a long, extensive, and inconclusive “diagnostic odyssey”.

The availability of a long follow-up for phenotype re-evaluation by a multidisciplinary team played a valuable crucial role in interpreting genetic findings; besides, it made it possible to confer and communicate intelligibility and coherence to parents in an often distressing developmental history [83]. Moreover, reaching a molecular diagnosis allowed posttest genetic counseling, determining the correct recurrent risk for parents and siblings, and deciding the proper reproductive strategies.

## 5. Conclusions

The present study demonstrates once more how rare and severe EEs, DEE_S,_ and NDDs have become more susceptible to diagnosis, with relevant significance for substantial approach and treatment options even in undiagnosed older and adult patients [84]. The high diagnostic yield reported must be interpreted in the particular research context of a tertiary health center. Nevertheless, it represents further evidence of the crucial role of reverse phenotyping and of collaboration between clinicians, basic scientists, and clinical researchers in the “exome first” approach. These factors will allow the deepening of mutual knowledge of this ever-expanding field in order to recognize diseases that can be treated with precision drugs in an earlier way.

## Figures and Tables

**Figure 1 genes-13-00500-f001:**
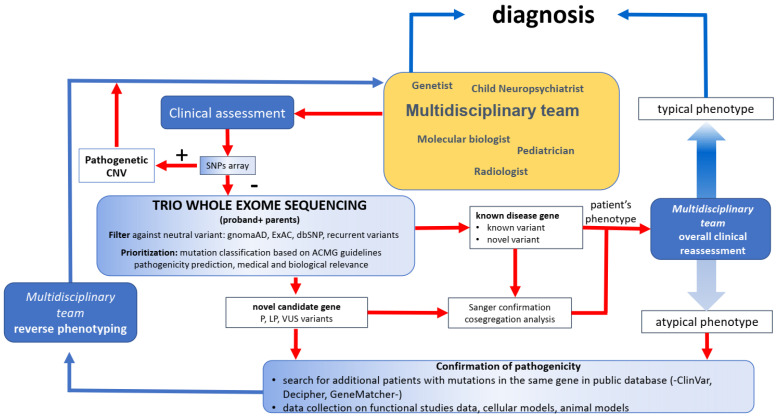
Schematic of the diagnostic procedure used in this study. The flow starts with the multidisciplinary team (yellow box) and goes to clinical assessment, molecular tests, and Sanger confirmations and cosegregation analysis (red arrows) and reaches the ending point (diagnosis) through a confirmation path (blue arrows) which includes clinical reassessment using a reverse phenotyping strategy. Abbreviations: SNP: single nucleotide polymorphism; CNV: copy number variation; gnomAD: the Genome Aggregation Database; ExAC: Exome Aggregation Consortium; dbSNP: the Single Nucleotide Polymorphism Database; ACMG: the American College of Medical Genetics and Genomics; P: pathogenic; LP: likely pathogenic; VUS: variant of unknown significance.

**Table 1 genes-13-00500-t001:** Characteristics of patients having whole-exome sequencing (WES).

Class of Age (Years)	Patient, *n*	Male, *n*	Female, *n*	Mean Age at the Time of WES (Years ± SD)
2–5	1	0	1	4
6–14	5	3	2	9.0 (±1.5)
15–22	11	6	5	18.5 (±2.4)
Total (2–22)	17	9	8	14.8 (±5.5)

**Table 2 genes-13-00500-t002:** Main clinical features of patients having WES.

Features	Percentage (%)
Epilepsy	100
DD/ID	88
Speech impairment	82
Refractory seizures	59
Microcephaly	24
Autism	18
Abnormal muscle tone	53
Dental abnormalities	12

**Table 3 genes-13-00500-t003:** The overall molecular diagnosis rate.

Diagnosis	Total Number of Patients (*n* = 17)	
	*n* of Patients	Patients’ Rate (%)
definite	10	58.8
possible	1	5.9
uncertain (VUS)	2	11.8
nondiagnosis	4	23.5

**Table 4 genes-13-00500-t004:** Definite genetic diagnosis (patients 1–10) or possible diagnosis (patient 11).

Patient-ID	Gender	Gene	Status	Variants	Known or Novel	Inheritance	ACMG	OMIM
1	M	*SLC13A5* [39]	cht	NM_177550: c.C1421T (p.P474L); c.G655A (p.G219R) chr17 (GRCh38): g.6690795G > A; g.6703031C > T	N; K	AR	LP; P	Epileptic encephalopathy, early infantile, 25 (MIM 615905)
2	F	*ROGDI*	hm	NM_024589: c.C286T (p.Q96X) chr16 (GRCh38): g.4800548G > A	K	AR	P	Kohlschütter–Tonz syndrome (MIM 226750)
3	M	*CACNA1H*	cht	NM_021098: c.G3175T (p.A1059S); c.C2329T (p.R777C) chr16 (GRCh38): g.1208033G > T; g.1204336C > T	K; N	AR	VUS; VUS	{Epilepsy, idiopathic generalized, susceptibility to, 6} MIM 611942; {Epilepsy, childhood absence, susceptibility to, 6} MIM 611942
4	F	*SCN1A*	cht	NM_001165963: c.C3521G (p.T1174S) c.C5782G (p.R1928G); chr2 (GRCh38): g.166015636G > C; g.165991493G > C	K; K	AR	LP; VUS	Epilepsy, generalized, with febrile seizures plus, type 2 (MIM 604403); Epileptic encephalopathy, early infantile, 6 (Dravet syndrome)(MIM 607208); Febrile seizures, familial, 3A (MIM 604403); Migraine, familial hemiplegic, 3 (MIM 609634)
5	M	*CDKL5*	ht	NM_003159: c.100-1G > A chrX (GRCh38): g.18564476G > A	N	somatic	P	Epileptic encephalopathy, early infantile, 2 (MIM 300672)
6	F	*MORC2*	ht	NM_001303256: c.G79A (p.E27K) chr22 (GRCh38): g.30958684 C > T	K	*de novo*	P	Developmental delay, impaired growth, dysmorphic facies, and axonal neuropathy (MIM 619090)
7	F	*GNB1*	ht	NM_002074: c.T239C (p.I80T) chr1 (GRCh38): g.1806503A > G	K	*de novo*	P	Mental retardation, autosomal dominant 42 (MIM 616973)
8	M	*SPATA5* [40]	hm	NM_145207: c.A1942G (p.K648E) chr4 (GRCh38): g.123028258A > G	N	AR	LP	Epilepsy, hearing loss, and mental retardation syndrome (MIM 616577)
9	M	*SPEN* [41]	ht	NM_015001:c.C3508T (p.R1170X) chr1(GRCh38): g.15929748 C > T	N	*de novo*	P	Radio-Tartaglia syndrome (MIM 619312)
10	F	*CSNK2B* [42]	ht	NM_001320: c.T116G (p.L39R) chr6 (GRCh38): g.31667911T > G	N	*de novo*	P	Poirier–Bienvenu neurodevelopmental syndrome (MIM 618732)
11	M	*SACS*	cht	NM_014363: c.G2983T (p.V995F); c.C3427A (p.Q1143K) chr13 (GRCh38): g.23340893C > A; g.23340449G > T	K; K	AR	VUS; P	Spastic ataxia, Charlevoix–Saguenay type (MIM 270550)

Abbreviations: M: male; F: female; AD: autosomal dominant; AR: autosomal recessive; cht: compound heterozygous; hm: homozygous; ht: heterozygous; sm: somatic; K: known; N: novel; VUS: variant of uncertain significance; P: pathogenic; LP: likely pathogenic.

## Data Availability

The data presented in this study are available upon request from the corresponding author. The data are not publicly available due to privacy restrictions.

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
