# Peer review of "The Genetic Diagnosis of Ultrarare DEEs: An Ongoing Challenge"

_genes, 2022, doi:10.3390/genes13030500_

Round 1

Reviewer 1 Report

This is a nicely designed and explained paper showing some important points about the evolution of genetic diagnosis of neurodevelopmental disorders, mainly the reanalysis of negative or inconclusive cases, the relevance of reverse phenotyping and the significance of homocygousity and compound heterocygosity.

The paper adds to the literature some those previous points. There is a concern about CACNA1H, which have been previously questioned as cause of epilepsy or encephalopathy:

Calhoun JD, Huffman AM, Bellinski I, Kinsley L, Bachman E, Gerard E, Kearney JA, Carvill GL. CACNA1H variants are not a cause of monogenic epilepsy. Hum Mutat. 2020 Jun;41(6):1138-1144. doi: 10.1002/humu.24017. Epub 2020 Apr 14. PMID: 32227660; PMCID: PMC7301766.

This has to be highlighted and maybe remove this patient, as one of the variants is a VUS (or even benign according to some platforms).

I would suggest to include also about the different phenotypes and some relevant issues of these techniques in adult patients: Beltrán-Corbellini Á, Aledo-Serrano Á, Møller RS, Pérez-Palma E, García-Morales I, Toledano R and Gil-Nagel A (2022) Epilepsy Genetics and Precision Medicine in Adults: A New Landscape for Developmental and Epileptic Encephalopathies. Front. Neurol. 13:777115. doi: 10.3389/fneur.2022.777115

Also, in this evolving strategy, I would include this new approach of "exome first": Klau J, Abou Jamra R, Radtke M, Oppermann H, Lemke JR, Beblo S, Popp B. Exome first approach to reduce diagnostic costs and time - retrospective analysis of 111 individuals with rare neurodevelopmental disorders. Eur J Hum Genet. 2022 Jan;30(1):117-125. doi: 10.1038/s41431-021-00981-z. Epub 2021 Oct 25. PMID: 34690354; PMCID: PMC8738730.

Check out for the use of new epilepsy nomenclature: better to use "focal" instead of "partial". Better to use "antiseizure medication" instead of "antiepileptic drug".

Check out for typos: elettroclinical (line 248).

Author Response

Response to Reviewer 1 Comments

Point 1: There is a concern about CACNA1H, which have been previously questioned as cause of epilepsy or encephalopathy:

Calhoun JD, Huffman AM, Bellinski I, Kinsley L, Bachman E, Gerard E, Kearney JA, Carvill GL. CACNA1H variants are not a cause of monogenic epilepsy. Hum Mutat. 2020 Jun;41(6):1138-1144. doi: 10.1002/humu.24017. Epub 2020 Apr 14. PMID: 32227660; PMCID: PMC7301766.

This has to be highlighted and maybe remove this patient, as one of the variants is a VUS (or even benign according to some platforms).

Response 1: We thank the reviewer for the comments regarding the CACNA1H gene and we certainly agree that heterozygous mutations are not a cause of monogenic epilepsy (and we have highlighted it in the text). Still, we believe that it is very important to present this case to discuss some issues (subsection 4.3 of the revisited manuscript) that could be of general interest for the scientific community: a) the CACNA1H gene plays an essential role in neuronal excitability and s spike-and-wave discharges that occur during absence seizures (PMID: 31217264); b) animal models carrying Cacna1h mutations co-segregate with epilepsy phenotypes (PMID: 32638069); c)The Ala1059Ser variant has been already reported to alter the channel activity (PMID: 17696120); d) The Ala1059Ser variant in compound heterozygous state with a second CACNA1H missense change (both inherited) have been recently reported in a patient with a complex neurological phenotype, underlying an additive effect of both variants on the channel activity (PMID: 26706850); e) Inherited variant in this gene could modify the clinical presentation in association with variants in other genes (PMID: 34399820).

Point 2: I would suggest to include also about the different phenotypes and some relevant issues of these techniques in adult patients: Beltrán-Corbellini Á, Aledo-Serrano Á, Møller RS, Pérez-Palma E, García-Morales I, Toledano R and Gil-Nagel A (2022) Epilepsy Genetics and Precision Medicine in Adults: A New Landscape for Developmental and Epileptic Encephalopathies. Front. Neurol. 13:777115. doi: 10.3389/fneur.2022.777115

Also, in this evolving strategy, I would include this new approach of "exome first": Klau J, Abou Jamra R, Radtke M, Oppermann H, Lemke JR, Beblo S, Popp B. Exome first approach to reduce diagnostic costs and time - retrospective analysis of 111 individuals with rare neurodevelopmental disorders. Eur J Hum Genet. 2022 Jan;30(1):117-125. doi: 10.1038/s41431-021-00981-z. Epub 2021 Oct 25. PMID: 34690354; PMCID: PMC8738730.

Response 2: We thank the Reviewer for the suggestion. The indicated references are very interesting and relevant. We have included them.

Point 3: Check out for the use of new epilepsy nomenclature: better to use "focal" instead of "partial". Better to use "antiseizure medication" instead of "antiepileptic drug".

Response 3: We have made changes according to the new epilepsy nomenclature.

Point 4: Check out for typos: elettroclinical (line 248).

Response 4: The manuscript has undergone to professional extensive English revision.

Reviewer 2 Report

The authors present the article entitled “The genetic diagnosis of ultra-rare DEEs: never give up!”. The article in its current form is hard to read and is not possible to extend my recommendation according to the following concerns:

Avoid using first-person sentences. Use instead third-person or passive voice sentences.

What is the intention to add the phrase “never give up!” in the Title? I suggest changing the title to be more attractive.

The abstract section must be restructured. This section should give a pertinent overview of the work. Avoid using headings in a single paragraph. I encourage the authors to read carefully the instructions for the authors.

Lines 62-73: These sentences are mentioning the same idea.

Introduction section must be carefully reviewed: Please give a pertinent background about the current state of the research field should be reviewed carefully and cite key publications.

The objective of the work is not clear. I suggest placing the objective by highlighting the novelty of the 

Please, at the end of the introduction include the structure of the manuscript.

Subsections must be numbered.

Before starting with this section, “Study design and participants” please integrate a paragraph to introduce the full section.

Line 249 can be supported by the three following references:

Hyperconnected Openings Codified in a Max Tree Structure: An Application for Skull-Stripping in Brain MRI T1; Application of morphological connected openings and levelings on magnetic resonance images of the brain; Sequential application of viscous opening and lower leveling for three-dimensional brain extraction on magnetic resonance imaging T1.

The images are very poor, vectorize them, use formal colors, can you define the meaning of the acronym of all the images like F1? ACMG, Exact, DBSNP….

The English should be improved, like this sentence that is kind of wire “To pinpoint a molecular diagnosis WES on trio (patient and both parents) was performed” (line 116).

Include a table that compares the findings of the work vs the already reported in the state of the art.

In general, this work is written in a slang mode, out of scientific writing. I recommend reformulating all the manuscript by starting in the title “...never give up”

Round 2

Reviewer 2 Report

The authors have addressed my comments.